# Performance Assessment of Different Precipitation Databases (Gridded Analyses and Reanalyses) for the New Brazilian Agricultural Frontier: SEALBA

Ewerton Hallan de Lima Silva [1], Fabrício Daniel dos Santos Silva [1,*], Rosiberto Salustiano da Silva Junior [1], David Duarte Cavalcante Pinto [1], Rafaela Lisboa Costa [1], Heliofábio Barros Gomes [1], Jório Bezerra Cabral Júnior [2], Ismael Guidson Farias de Freitas [3] and Dirceu Luís Herdies [4]

1   Institute of Atmospheric Sciences, Federal University of Alagoas, Maceió 57072-900, Brazil;
    ewerton.hallan@ifal.edu.br (E.H.d.L.S.); rosiberto@icat.ufal.br (R.S.d.S.J.); david.duarte@icat.ufal.br (D.D.C.P.);
    rafaela.costa@icat.ufal.br (R.L.C.); heliofabio@icat.ufal.br (H.B.G.)
2   Institute of Geography, Development and Environment, Federal University of Alagoas,
    Maceió 57072-900, Brazil; jorio.cabral@igdema.ufal.br
3   Academic Unit of Atmospheric Sciences, Center for Technology and Natural Resources,
    Federal University of Campina Grande, Campina Grande 58109-970, Brazil; ismael.freitas@icat.ufal.br
4   National Institute for Space Research, Cachoeira Paulista, São Paulo 12630-000, Brazil; dirceu.herdies@inpe.br
*   Correspondence: fabricio.santos@icat.ufal.br

**Abstract:** Since the early 2000s, Brazil has been one of the world's leading grain producers, with agribusiness accounting for around 28% of the Brazilian GDP in 2021. Substantial investments in research, coupled with the expansion of arable areas, owed to the advent of new agriculture frontiers, led the country to become the world's greatest producer of soybean. One of the newest agricultural frontiers to be emerging in Brazil is the one known as SEALBA, an acronym that refers to the three Brazilian states whose areas it is comprised of—Sergipe, Alagoas, and Bahia—all located in the Northeast region of the country. It is an extensive area with a favorable climate for the production of grains, including soybeans, with a rainy season that takes place in autumn/winter, unlike the Brazilian regions that are currently the main producers of these kinds of crops, in which the rainfall regime has the wet period concentrated in spring/summer. Considering that precipitation is the main determinant climatic factor for crops, the scarcity of weather stations in the SEALBA region poses an obstacle to an accurate evaluation of the actual feasibility of the region to a given crop. Therefore, the aim of this work was to carry out an assessment of the performance of four different precipitation databases of alternative sources to observations: two from gridded analyses, MERGE and CHIRPS, and the other two from ECMWF reanalyses, ERA5, and ERA5Land, and by comparing them to observational records from stations along the region. The analysis was based on a comparison with data from seven weather stations located in SEALBA, in the period 2001–2020, through three dexterity indices: the mean absolute error (*MAE*), the root mean squared errors (*RMSE*), and the coefficient of Pearson's correlation (*r*), showing that the gridded analyzes performed better than the reanalyses, with MERGE showing the highest correlations and the lowest errors (global average *r* between stations of 0.96, followed by CHIRPS with 0.85, ERA5Land with 0.83, and ERA5 with 0.70; average *MAE* 14.3 mm, followed by CHIRPS with 21.3 mm, ERA5Land with 42.1 mm and ERA5 with 50.1 mm; average *RMSE* between stations of 24.6 mm, followed by CHIRPS with 50.8 mm, ERA5Land with 62.3 mm and ERA5 with 71.4 mm). Since all databases provide up-to-date data, our findings indicate that, for any research that needs a complete daily precipitation dataset for the SEALBA region, preference should be given to use the data in the following order of priority: MERGE, CHIRPS, ERA5Land, and ERA5.

**Keywords:** climate; precipitation; reanalysis; grain; agribusiness; SEALBA

## 1. Introduction

Brazil has consolidated itself in the global agricultural scenario as one of the largest grain producers in the world, and still on the rise. In 2020, grain production (including cereals, legumes, and oilseeds) was 243.2 million tons, 2.2% higher than the 2019 harvest, which was 241.4 million tons [1]. These are significant numbers for a country that, until the 1970s, was a food importer, mainly due to the low technology used in the agricultural sector, little knowledge about its tropical soils and their response to fertilizer applications, inability to develop high-yield varieties adapted to these soils, inadequate management practices, and lack of agricultural policies aimed at the development of production and productivity [2].

This scenario changed from 1980, under the strong contribution of the Brazilian Agricultural Research Corporation (Embrapa, from its acronym in Portuguese), created in 1972, under the Ministry of Agriculture and with decentralized research centers throughout Brazil, initially focused on research on wheat, rice, and beans [3]. Today, Embrapa is present in all regions of the country, playing an important role in the consolidation of Brazilian agribusiness [4,5]. As a result of continued investments in agricultural research and development, rural extension, and credit for producers [6], Brazil became a global highlight in agribusiness from the 2000s onwards; as an example of this importance, in 2019 the GDP agribusiness represented 21.4% of the total Brazilian GDP [7]. Regarding soybeans, the country was the world's largest producer of the grain in the last crop 2020/21 with a production of 135,409 million tons, followed by the United States with 112,549 million tons.

With continental dimensions, soils, and varied climates, one of the main reasons for the expansion of Brazilian grain production was the adaptation of areas with no previous production history, called new agricultural frontiers. One of the most recent and successful is known as MATOPIBA (acronym for the Brazilian states where it is located: Maranhão, Tocantins, Piauí, and Bahia). Between 1996 and 2006 alone, soy production grew by more than 280% in MATOPIBA [8]. The increase in productivity in MATOPIBA is due, among other factors, to the extensive flat areas, high insolation, and well-defined seasonality of precipitation, concentrated between 4 and 5 months. However, the recent evolution of precision agriculture and the wealth brought by agribusiness with the consequent change in land use and occupation [9] also brought changes in climate [10–12]. In this sense, a new agricultural frontier with high potential for grain production appears in Brazil, called SEALBA (abbreviation of the Brazilian states that compose it: Sergipe (SE), Alagoas (AL), and Bahia (BA)).

Most of the agricultural production in SEALBA comes from family and subsistence agriculture, associated with low-income farmers who employ little or no technology, who mainly cultivate maize, beans, and cassava. Precipitation is the main limiting factor for the success/failure of production, directly responsible for the increase in soil moisture levels, affecting all arable areas on the planet, which can be monitored by indices that express the vegetative vigor [13–17]. At SEALBA [18], it was found that bean productivity is up to 70% lower than that observed in states in the southern region of Brazil; in this sense, the introduction of crops such as soybeans can bring considerable socioeconomic gains to local producers [19].

Detailed climatic knowledge in areas of intense agricultural production is fundamental for planning purposes. As observed in the studies of the MATOPIBA region, the lack of continuous, long-term, and flawless observed data becomes essential, and its poor spatio-temporal distribution needs to be addressed. Among the meteorological variables observed on the surface, precipitation is of greater importance, mainly for agriculture. There are few weather stations in SEALBA, so the purpose of this article is to identify the best precipitation database for SEALBA among four available products, two of grid analysis that mix surface observations and satellite estimates—MERGE [20–22] and CHIRPS [23–25]—and two reanalyses from the European Center for Medium-Range Weather Forecasts (ECMWF)—ERA5 and ERA5Land [26–29].

It is worth noting that Brazil continually suffers from the gradual closure of weather stations [30], and the comparison of observed data with data from gridded (re)analyses in order to assess their performance is a necessity not only in Brazil, but at a global level [31,32]. In view of the increasingly present perspective of conducting climate studies and generating products for decision-making based on these sources of information, the need for this study is justified once again for an area of Brazil that may soon consolidate it as the world's largest producer of grains. Furthermore, it is intended that such analyzes help in the validation of products from the surface observation database of the Brazilian Global Atmospheric Model (BAM) [33], which is the atmospheric module of the Brazilian Earth System Model (BESM), aiming to obtain a hybrid dynamic-statistical coupling for the observed surface data and make adjustments in seasonal forecast products from BAM to NEB, with emphasis on SEALBA.

## 2. Materials and Methods

### 2.1. Region of Study

SEALBA is a continuous and interconnected area of municipalities in three states in the eastern part of the NEB, a region historically linked to sugarcane production since the beginning of European occupation in Brazil. According to [34], 33.2% of the area in the region is in Sergipe (1,707,815 ha), 36.1% in Alagoas (1,859,438 ha), and 30.7% in Bahia (1,581,688 ha), totaling 5,148,941 hectares. SEALBA is made up of 171 municipalities, 69 of which are located in Sergipe, 74 in Alagoas, and 28 in northeastern Bahia (Figure 1).

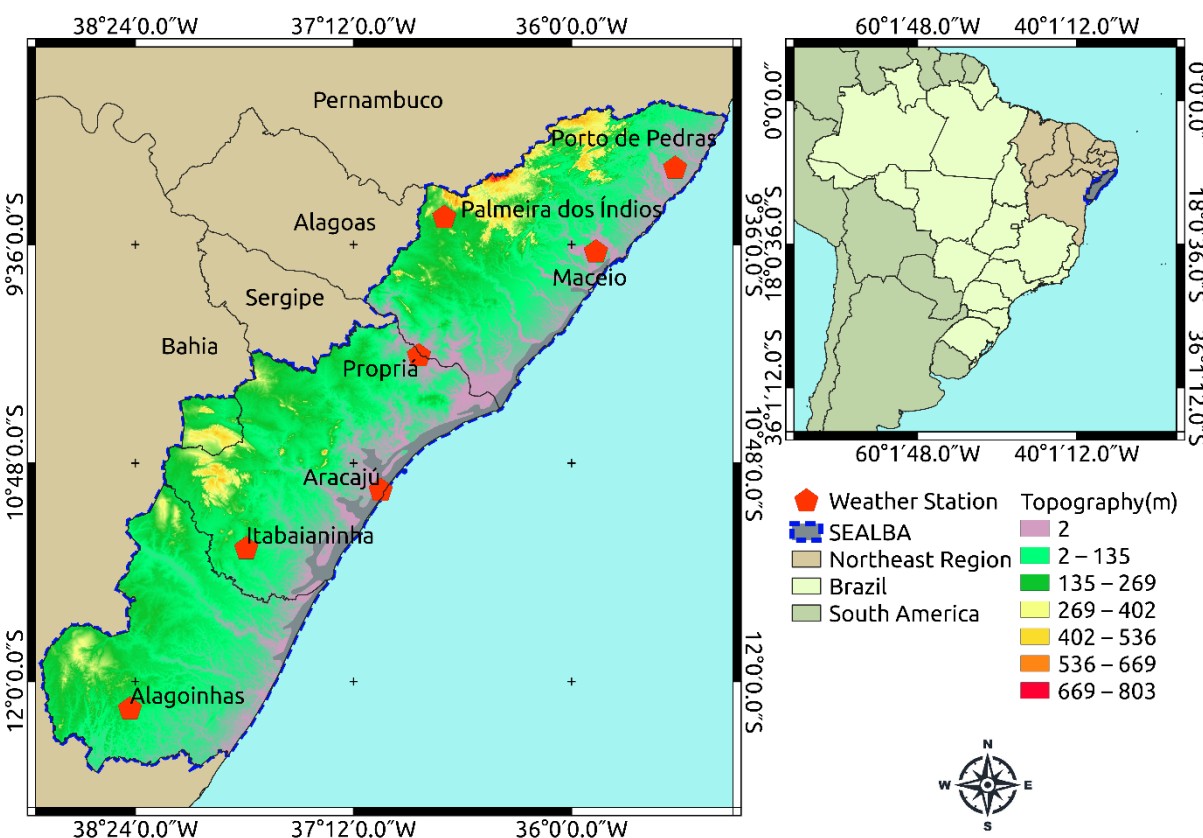

**Figure 1.** SEALBA map with highlighted topography, and proportion of its area in relation to NEB (beige), Brazil (yellow), and South America (grey). The INMET weather stations whose data were used in this study are identified by red points.

Figure 1 shows the location of SEALBA and its area in relation to the Brazilian territorial extension. It is an area of low relief, with some mountain areas that exceed 400 m in height. In most of the region, altitudes barely rise above mean sea level. SEALBA's rainy season is in autumn/winter, and it has a large number of perennial/semiperennial rivers

that bathe this portion of the eastern part of the NEB. SEALBA has 32% of its territory located on the border of the Brazilian semi-arid region, with lower rainfall than in the extreme east. Among the advantages of this region are the proximity of two ports, located in Alagoas and Sergipe, in addition to two other ports in Bahia, which brings advantages for the flow of agricultural production in this region.

In the entire SEALBA, there are only seven conventional/manual weather stations with historical data from 1961 operated by the National Institute of Meteorology (INMET), three in the SEALBA portion of Alagoas (AL), three in Sergipe (SE), and one in Bahia (BA), whose metadata are described in Table 1.

**Table 1.** Metadata of the conventional weather stations located in the SEALBA region. Source: INMET.

| Municipality with Weather Station | State | Latitude (°) | Longitude (°) | Altitude (m) |
| --- | --- | --- | --- | --- |
| Maceió | AL | −9.67 | −35.70 | 64.50 |
| Palmeira dos Índios | AL | −9.45 | −36.70 | 274.90 |
| Porto de Pedras | AL | −9.18 | −35.43 | 50.02 |
| Aracajú | SE | −10.95 | −37.05 | 4.72 |
| Itabaianinha | SE | −11.27 | −37.79 | 208.00 |
| Propriá | SE | −10.21 | −36.84 | 19.92 |
| Alagoinhas | BA | −12.15 | −38.43 | 130.92 |

### 2.2. Gridded Analysis Data

For any activity that requires precipitation data, continuity is essential, with the guarantee of constant updating of the time series. In this sense, we selected for analysis two sources of precipitation data, the MERGE developed in Brazil by the *Instituto de Pesquisas Espaciais* (INPE), and the CHIRPS, developed and made available by the United States Geological Survey (USGS). Each of the sources is better described below.

### 2.2.1. MERGE

MERGE [20,22] consists of combining data observed in precipitation networks operated by different Brazilian agencies with data from the GPM-IMERG-EARLY satellite [35], after the discontinuation of the TRMM-TMPA [36]. MERGE data are made available by CPTEC/INPE in daily format for all of South America with a resolution of 10 km. Daily MERGE data are available since 1 January 2000, at: http://ftp.cptec.inpe.br/modelos/tempo/MERGE/GPM/DAILY/ (accessed on 31 January 2022).

### 2.2.2. CHIRPS

The CHIRPS data comes from efforts by US institutions to map precipitation, especially in areas with scarce surface observations and in areas of complex terrain, such as mountain ranges [23]. CHIRPS uses all the data from a network of surface observations to construct a high-resolution mesh, together with precipitation estimates based on satellite observations in the infrared band from an algorithm based on cold cloud duration [37]. The satellite-generated data are then combined with the World Meteorological Organization (WMO) Global Telecommunications System (GTS) surface observations and resampled on a 0.05° grid, taking into account the physiographic characteristics of the surface. CHIRPS data is available in different resolutions, 0.25° × 0.25° and 0.05° × 0.05°, from 1981 to the present, at: https://data.chc.ucsb.edu/products/CHIRPS-2.0/ (accessed on 31 January 2022).

### 2.3. Reanalysis Data

Another alternative to provide data to areas with a network of scarce surface observations is the use of reanalysis. Saurral et al. [38] and, more recently [30] show that all of South America, and especially Brazil, suffered a significant reduction in the number of weather stations, many being definitively closed, which forces the use from alternative sources of data such as those from reanalysis, which provide a synthetic database reconstructed for

previous periods, based on the calibration of climate models. These data need to be analyzed from the comparison with historical observations and need to demonstrate statistical parameters similar to those of the observations, such as means and variances [39]. In this research, the two most recent versions of the ECMWF reanalyses, ERA5 and ERA5Land, briefly described below, were evaluated.

ERA5 and ERA5Land

ERA5 is the fifth generation of atmospheric reanalyses produced by the ECMWF, replacing the previous ERA-Interim [40,41] and ERA-40 [42]. Its coverage period is from 1950 to the present, with hourly outputs on a grid with a spatial resolution of 31 km. ERA5 uses the 4D-Var data assimilation scheme based on the Integrated Forecasting System (IFS) Cy41r2, operational since 2016. In a study by [43], an increase in the global average correlation of precipitation of 10% was observed compared to Global Precipitation Climatology Project (GPCP) data. ERA5Land represents an evolution of ERA5, describing in greater detail the water and energy cycles in a smaller grid spacing of 9 km, and also with an hourly temporal frequency [29]. Preliminary evaluations showed higher quality of ERA5Land in the representation of temperature at 2 m, soil moisture, and river runoff. However, although improvements in the representation of surface precipitation can be inferred, this is not a guarantee, and evaluating its performance in different locations around the globe under different climatic conditions, such as in the SEALBA region, is necessary.

*2.4. Methodology for Intercomparison*

Due to the low density of stations and the impossibility of extrapolating the data from this network, the comparison between observed data and each grid analysis/reanalysis is based on extracting the synthetic time series of each base from the same geographic coordinates of the stations. As the spatial resolutions are varied, and any grid will always provide four points around a point of interest (weather station), each synthetic series was extracted using the simple bilinear interpolation method [44]. This method calculates a variable value at a specific grid point, assigning characteristic weights to each of the four grid points in relation to the actual location of the weather station, with greater weights the closer the grid point is to the point of interest [45]. This quantitative verification of the precipitation grid point $\times$ grid point and/or grid point $\times$ actual observation is used in many studies that have attested to its effectiveness [36,46,47]. The mean absolute error (*MAE*, Equation (1)) and the mean square error (*RMSE*, Equation (2)) were used to assess the accuracy of the databases [48–50]. Pearson's correlation coefficient (*r*, Equation (3)) was used to identify the variability relationship between observed precipitation $\times$ precipitation from the (re)analysis [51]. All statistical analyzes were performed using the free software R in its version 4.0.3.

$$MAE = \frac{1}{N} \sum_{i=1}^{N} ABS(P_i - O_i) \tag{1}$$

$$RMSE = \sqrt{\frac{1}{N} \sum_{i=1}^{N} (P_i - O_i)^2} \tag{2}$$

$$r = \frac{Cov(o, p)}{\sigma(o, p)} \tag{3}$$

where $N$ is the total number of elements in the series, $P_i$ = precipitation of the gridded analysis/reanalysis, $O_i$ = observations at each time $i$, $Cov(o, p)$ is the covariance between the data, and $\sigma(o, p)$ are the respective standard deviations.

To ensure that the value of $r$ really expresses the agreement between observations and (re)analyses, the parametric Student's t-test [52] was used to verify the statistical significance of correlations at a confidence interval of 99% (*p*-value < 0.01). A premise for

using the test is that the sample size, *N*, from which the value of the correlation coefficient, *r*, is obtained is equal to or greater than 6, then the value of t is given by Equation (3):

$$t = \frac{r}{\sqrt{\frac{(1-r^2)}{(N-2)}}} \tag{4}$$

Equation (4) is a distribution for *t* with $N - 2$ degrees of freedom. Applying this formula to any value of *r* and *N* will test the null hypothesis that the observed value comes from a population in which there is no significant correlation between the data. Once the value of t is obtained, the critical correlation coefficient ($r_c$) can be extracted, a value for which the statistical hypothesis that there is a correlation between simulated and observed data is accepted or not, $r_c$ is given by Equation (4):

$$r_c = \sqrt{\frac{t^2}{(N-2) + t^2}} \tag{5}$$

As the data periods differ between sources, 1961 to the present (INMET), 2000 to the present (MERGE), 1981 to the present (CHIRPS), and 1979 (1950) to the present (ERA5 and ERA5Land), all comparative analyzes were performed for the period in common among all data sources: 2001 to 2020, always obtaining as a reference variable the precipitation observed in the rain gauge of the INMET weather station.

## 3. Results and Discussions

### 3.1. Precipitation Regimes

Figure 2 shows the average annual cycle of precipitation in the period 2001–2020 of the seven INMET conventional weather stations at SEALBA, which constitutes an area that extends from the coastal strip to the border with the northeastern semi-arid region. Of these, three are located on the coast: Porto de Pedras, Maceió, and Aracajú, which concentrate the highest amounts of precipitation, and four in the transition area between the coast and the semi-arid region, known as agreste: Propriá, Itabaianinha, Alagoinhas, and Palmeira dos Índios, with lower annual average accumulated, marking an expressive East-West zonal gradient of precipitation also described in [53].

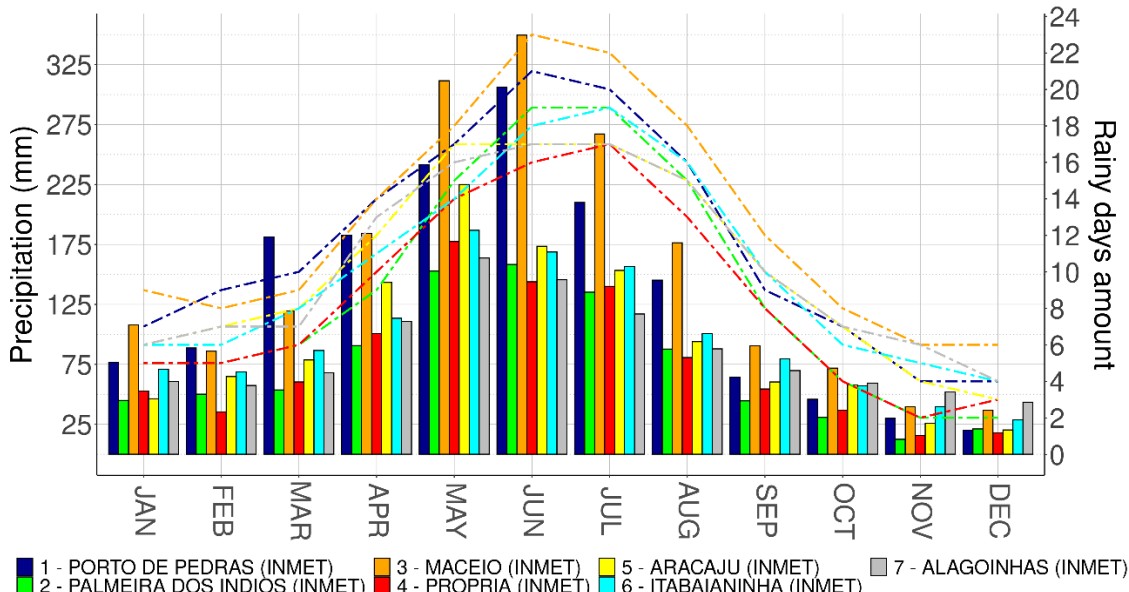

**Figure 2.** Monthly precipitation measured at the SEALBA stations averaged over the period 2000–2020 (columns, left axis) and number of days with precipitation greater than 1 mm (dashed lines, right axis).

The wettest semester, from March to August, concentrates approximately 75% of the total annual precipitation. June is the wettest month in the north of SEALBA (Porto de Pedras and Maceió), and May in the other locations, with the May–June–July quarter being the wettest of the year in all weather stations, characteristic of the climate classified as *As* [54], whose maximum precipitation occurs in autumn-winter, which are also the months of the year with the highest number of rainy days. SEALBA has as climatic advantages the occurrence of precipitation in volumes greater than 450 mm from March to August in at least 50% of the total area of its municipalities, a volume of precipitation sufficient for the cultivation of several grain crops, in addition to the component temperature that is essential to reach the maximum production achievable by a crop [19,55].

In the driest quarters, the accumulated precipitation varies between 50 and 250 mm, in municipalities located at the western end of the SEALBA. This same area, in the wettest quarters, has accumulations that vary on average from 200 to just over 300 mm [11].

The main rain-causing system consists of the easterly wave disturbances, which are large masses of clouds that move through the Atlantic Ocean and over the continent up to 200 km inland, causing intense rains on the coast, with gradually lower volumes as they advance towards the interior [56,57]. Breeze circulation is one of the main regulating mechanisms of the precipitation regime in eastern NEB, whose intensity is modulated by large-scale circulation and topography and land use effects [58–61]. Other important atmospheric systems for precipitation in SEALBA are upper tropospheric cyclonic vortices, frontal systems, and mesoscale convective complexes [62–73].

### 3.2. Comparison between Points and Stations in the Monthly Time Scale

Table 2 shows the results of comparing the monthly rainfall observed in each weather station and obtained by MERGE, CHIRPS, ERA5, and ERA5Land, between 2001 and 2020. Due to the sample size, $N = 240$, the critical correlation $r_c$ at a confidence level of 99.9% by the Student's t-test is 0.22, that is, any $r_c \geq 0.22$ indicates that there is a significant correlation between the observed data and the (re)analysis. Correlations ranged from 0.62 to 0.99, all of which were statistically significant (*p*-value < 0.001). Based on *r* values, MERGE had the best performance in six of the seven stations, with the exception of Aracajú station, where MERGE presented $r = 0.93$ and CHIRPS $r = 0.95$, the only one where MERGE was surpassed by another base of data, although with a very small difference when compared to other locations where MERGE outperformed CHIRPS. From the global correlation between stations, MERGE presented the best performance among the sources, with $r = 0.96$, followed by CHIRPS with $r = 0.85$, ERA5Land with $r = 0.83$ and ERA5 with $r = 0.70$.

The *RMSE* was selected as a dexterity estimation measure because it has, among other advantages, the possibility of expressing the accuracy of numerical results with error values in the same dimensions of the analyzed variables [51], that is, millimeters for the precipitation. In this sense, the data source that presented the best performance in relation to the observations was CHIRPS, with a global average error between the stations of 24.6 mm, followed by MERGE with 50.8 mm, ERA5 with 71.4 mm, and ERA5Land with 82.2 mm.

The *MAE* was also selected to measure the average magnitude of the errors of the (re)analysis in relation to the observations, regardless of their direction, considering the absolute differences between the data. In this sense, it is a more interesting measure than bias, where positive and negative errors tend to cancel each other out in unbiased samples. In this sense, the *MAE* values obtained corroborate those of the *RMSE*, placing the MERGE in order of greater to lesser dexterity, with a global *MAE* between stations of 14.3 mm, CHIRPS with 32.3 mm, ERA5Land with 42.1 mm, and ERA5 with 50.1 mm.T

**Table 2.** Statistical analysis (Pearson correlation *r*, *RMSE*, and *MAE*) on the monthly-accumulated precipitation from the different databases versus observations in the 2001–2020 period. The highest correlations are highlighted in bold.

| Weather Station | Index | Data Base | | | |
| --- | --- | --- | --- | --- | --- |
| | | MERGE | CHIRPS | ERA5 | ERA5 LAND |
| Porto de Pedras/AL | *r* | **0.96** *** | 0.72 *** | 0.71 *** | 0.85 *** |
| | *RMSE* | 34.6 | 92.8 | 107.9 | 85.0 |
| | *MAE* | 20.0 | 53.1 | 72.1 | 56.7 |
| Palmeiras dos Índios/AL | *r* | **0.95** *** | 0.81 *** | 0.72 *** | 0.82 *** |
| | *RMSE* | 24.0 | 46.4 | 49.9 | 45.6 |
| | *MAE* | 12.1 | 29.9 | 38.1 | 30.6 |
| Maceió/AL | *r* | **0.99** *** | 0.85 *** | 0.81 *** | 0.9 *** |
| | *RMSE* | 24.5 | 71.5 | 108.3 | 91.2 |
| | *MAE* | 13.8 | 43.9 | 67.7 | 59.9 |
| Propriá/SE | *r* | **0.97** *** | 0.89 *** | 0.72 *** | 0.78 *** |
| | *RMSE* | 17.4 | 35.9 | 52.2 | 56.4 |
| | *MAE* | 9.3 | 22.7 | 39.5 | 38.2 |
| Aracajú/SE | *r* | 0.93 *** | **0.95** *** | 0.67 *** | 0.85 *** |
| | *RMSE* | 31.8 | 31.1 | 70.1 | 52.0 |
| | *MAE* | 20.27 | 22.22 | 47.77 | 34.82 |
| Itabaianinha/SE | *r* | **0.96** *** | 0.85 *** | 0.64 *** | 0.82 *** |
| | *RMSE* | 23.5 | 44.7 | 58.9 | 55.3 |
| | *MAE* | 15.1 | 30.1 | 43.7 | 37.9 |
| Alagoinhas/BA | *r* | **0.97** *** | 0.87 *** | 0.62 *** | 0.77 *** |
| | *RMSE* | 16.1 | 33.0 | 52.7 | 50.4 |
| | *MAE* | 9.8 | 24.4 | 41.5 | 36.8 |

*** indicates $p < 0.001$.

It is worth noting that the estimates made for precipitation based on satellite images calibrated with surface observations (MERGE and CHIRPS) were the most efficient compared to the estimates based on reanalysis data (ERA5Land and ERA5). However, this does not render the use of reanalysis data inadvisable, as these presented *r* values from 0.77 to 0.90 (ERA5Land) and from 0.62 to 0.81 (ERA5), which were statistically significant for monthly precipitation, with the ones relative to ERA5Land, more specifically, being very close to those obtained with CHIRPS. Moreover, the reanalysis data have some advantages that need to be pointed out, such as the availability of other meteorological variables: temperature, relative humidity, wind, radiation, and potential evapotranspiration [74]. Regarding the errors estimated by *RMSE* and *MAE*, the proximity of the values obtained by ERA5Land with those of CHIRPS is noticeable, with minimum and maximum *RMSE* obtained between stations ranging from 45.6 to 91.2 mm compared to 31.1 to 92.8 mm of CHIRPS, and with even smaller differences when comparing the minimum and maximum MAE obtained between the stations, which varied from 30.6 to 59.9 mm with ERA5Land versus 22.2 to 53.1 mm with CHIRPS.

Similar results were also found by [75], regarding the accuracy of the CHIRPS database, which showed systematic errors of underestimation (negative bias) and overestimation (positive bias) in the four seasons of the year in different SEALBA locations. On the other hand, *RMSE* values do not exceed 5 mm per season per year (considering all accumulated for summer, autumn, winter, and spring). [22] highlights the good representation of MERGE for the NEB coast (region R4 in their article) in terms of *r* and *RMSE* values and emphasizes the importance of having margins to further improve the precipitation estimates from a better representation in satellite images of the warm clouds that are usually responsible for causing precipitation along the eastern NEB.

The wettest period occurs between April and July; on the other hand, the driest period begins in September and lasts until December, throughout the SEALBA region. However, it should be noted that the amount of precipitation varies between different locations and within each location. A brief analysis for a coastal weather station, Maceió, and a weather station in the interior of SEALBA, Alagoinhas, was performed and the results presented in Figure 3. The results in boxplot format for the two locations (in yellow in the figures), show that MERGE (in violet) also has the best ability to represent extreme positive and negative values, both in winter and in summer, surpassing the other databases that underestimate/overestimate above-average precipitation in winter. The same occurs in the summer, with the tendency of the databases to overestimate the accumulated precipitation observed, on average, and fail to represent the extreme values well.

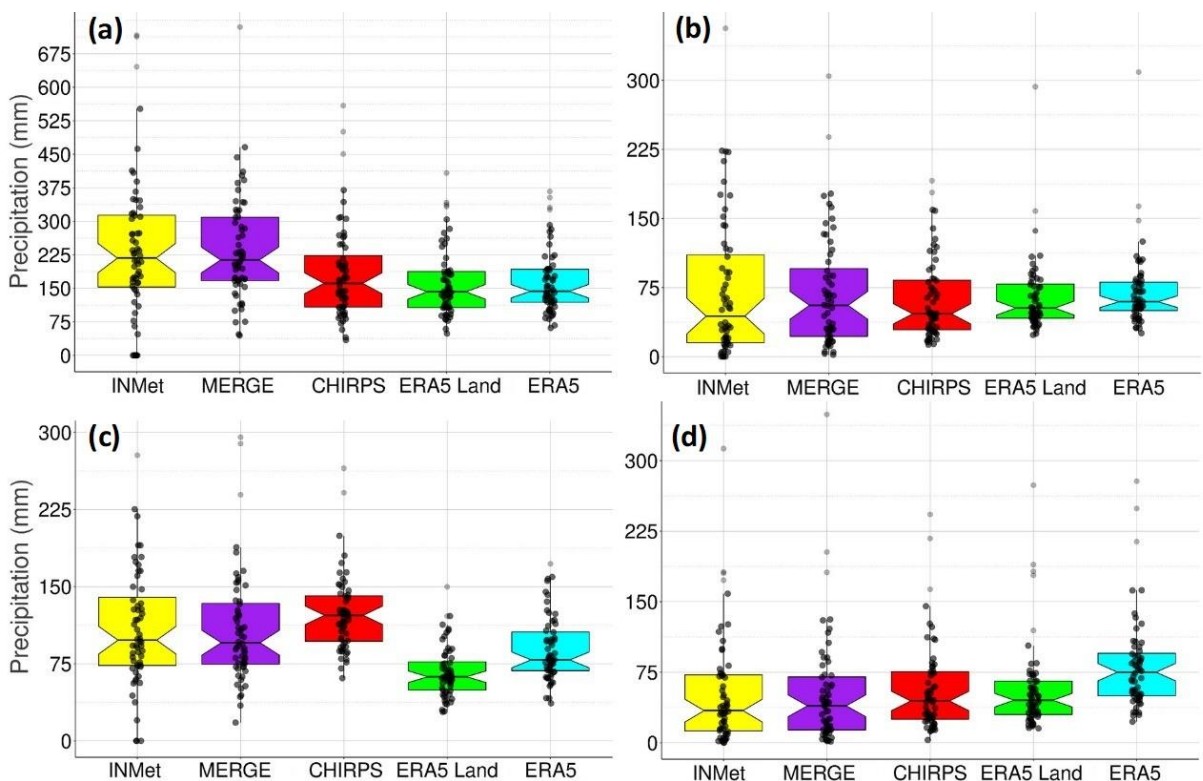

**Figure 3.** Comparison charts of the climate average between the observed precipitation (yellow) and data from MERGE (violet), CHIRPS (red), ERA5Land (green), and ERA5 (turquoise), for Maceió/AL (upper panel) in the winter (**a**) and in the summer (**b**) and for Alagoinhas/BA (lower panel) in the winter (**c**) and in the summer (**d**).

### 3.3. Intercomparison between Databases in SEALBA

The results presented point to MERGE as the best alternative among the databases analyzed in SEALBA, more reliable and accurate for applications that require continuous data of daily precipitation.

In this direction, Figure 4 shows the annual average monthly precipitation of each database, taking the MERGE as a reference (Figure 4a). The results show the strong horizontal gradient of precipitation. In the rainiest coastal strip, there are three well-defined areas: a rainier portion north of the SEALBA, in the state of Alagoas, where some municipalities exceed 150 mm in the annual monthly average; and two less rainy: on the border between Alagoas and Sergipe, and on the coast of Bahia, with annual monthly averages around 100 mm. In the municipalities on the western border of SEALBA, precipitation ranges from 50 to 75 mm in annual monthly averages.

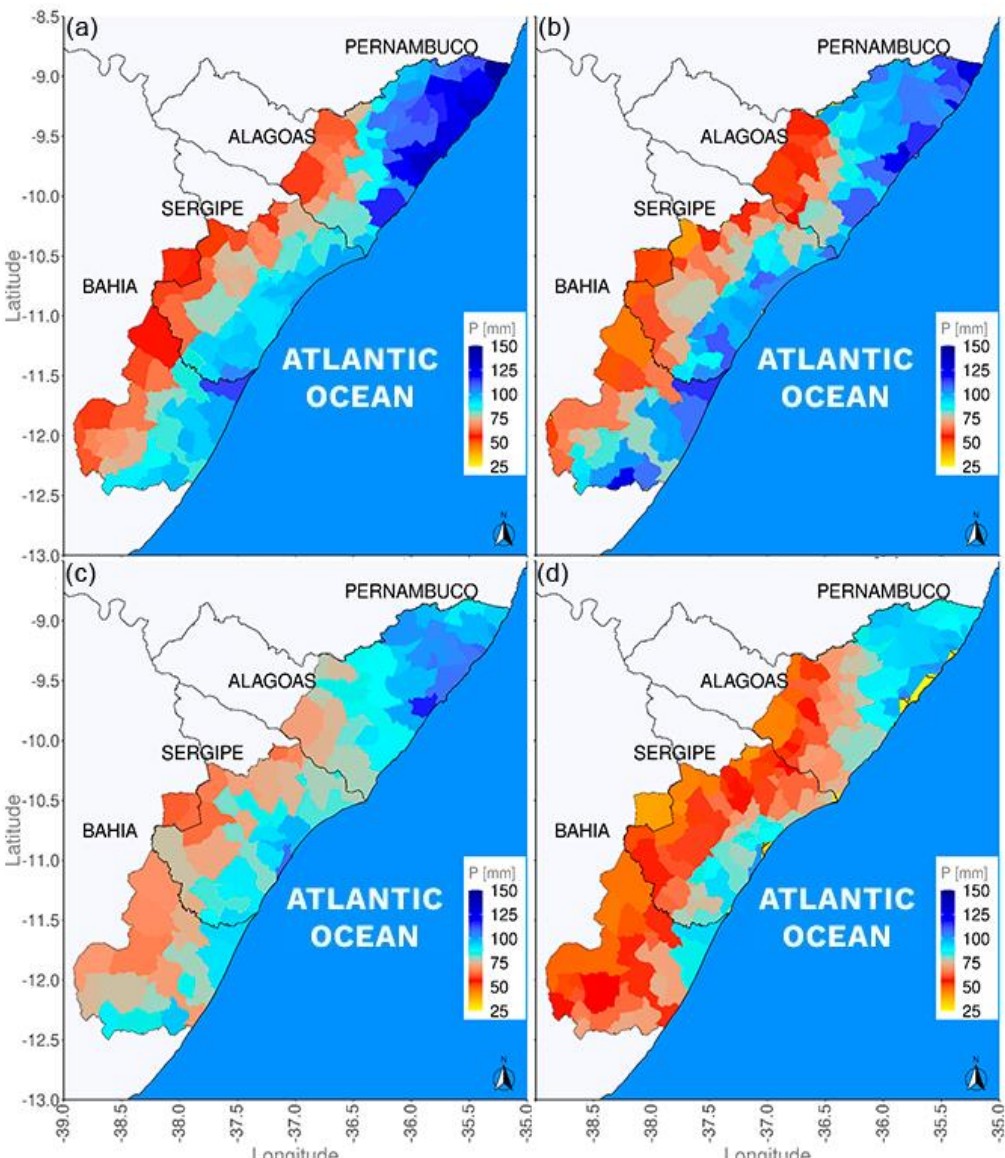

**Figure 4.** Annual monthly average of accumulated precipitation as per data from MERGE (**a**), CHIRPS (**b**), ERA5 (**c**), and ERA5Land (**d**). The precipitation scale on the map (P) varies from 0 to 150 mm.

Figure 5 shows the differences between MERGE (Figure 5a) and each database. Better results can be seen from CHIRPS (Figure 5b), with the smallest differences in relation to MERGE, followed by ERA5 (Figure 5c) and ERA5Land (Figure 5d), respectively, with the largest differences. CHIRPS, ERA5, and ERA5Land underestimate the precipitation in the north of the SEALBA, a common point between these three databases. CHIRPS overestimates precipitation in most municipalities in the center and south of SEALBA, and underestimates in most municipalities on the western border. ERA5 mostly underestimates precipitation in the eastern range and overestimates in the western range. Relative to MERGE, ERA5Land underestimates precipitation across SEALBA.

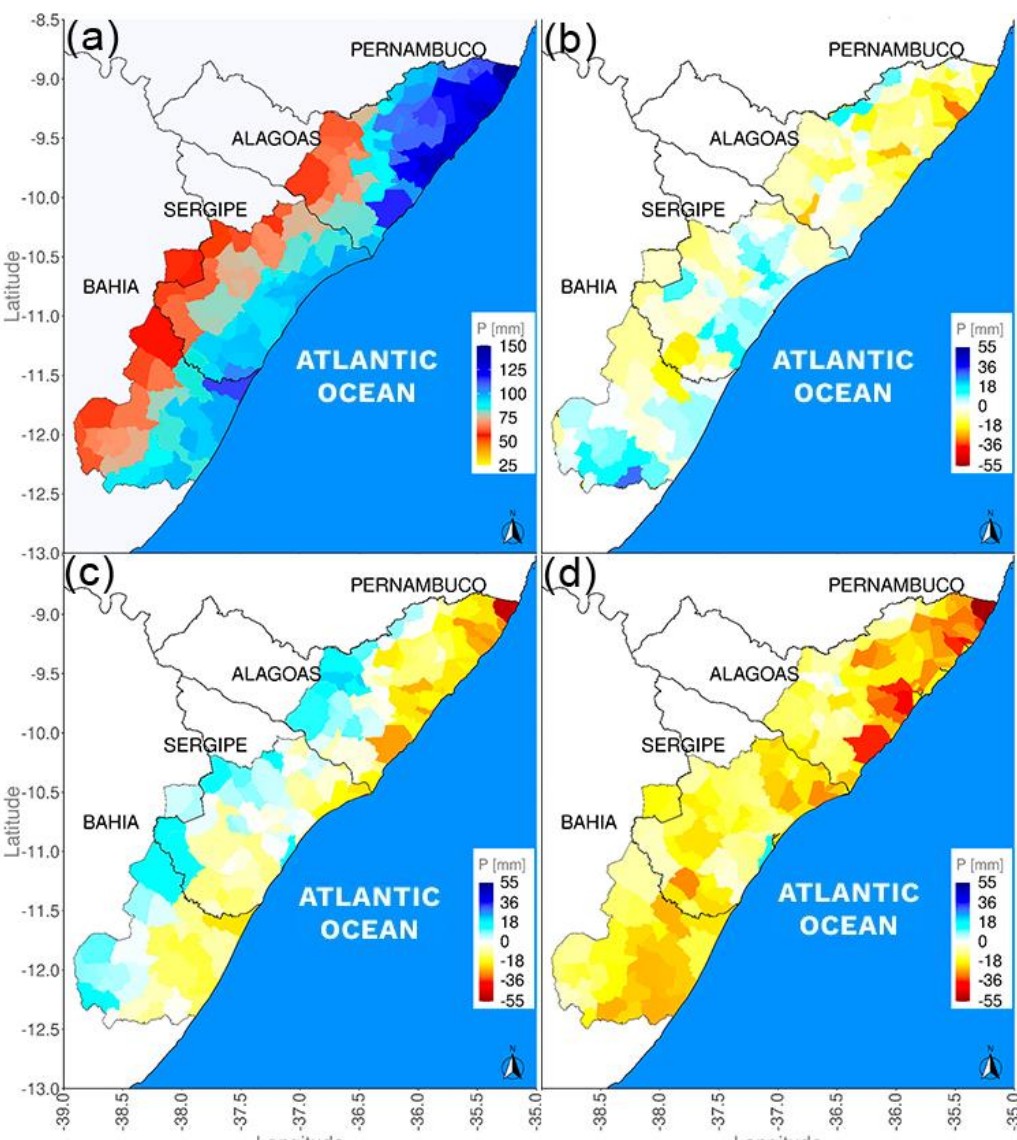

**Figure 5.** Annual monthly average of accumulated precipitation as per data from MERGE (**a**) and the differences between that and the corresponding values obtained from CHIRPS (MERGE minus CHIRPS) (**b**), ERA5 (MERGE minus ERA5) (**c**), and ERA5Land (MERGE minus ERA5Land) (**d**). The precipitation scale (P) in (**b**–**d**), varies from −55 to 55 mm.

In general, the results presented are in agreement with the literature. Ref. [75] compared the CHIRPS data with observations in the NEB and, in particular, in the SEALBA region, showing that this data source represents well the climatological pattern of precipitation. Ref. [24] validated the CHIRPS by comparing its monthly accruals with a small set of stations in the NEB. At SEALBA, the only common station used with our work was Aracajú, with the authors showing that in the period 1981–2013 CHIRPS tended to overestimate the observed precipitation, with a correlation of 0.96. In our analyses, even for a different period, from 2000 to 2020, Aracajú was the only station where CHIRPS presented a higher correlation than MERGE, 0.95 versus 0.93, but underestimating precipitation in relation to MERGE.

Rozante et al. [22], when analyzing the performance of two precipitation products generated by INPE from satellite images, MERGE and CoSch, reached the conclusion that even with the deficiency of the GPM-IMERG-EARLY satellite in representing the precipitation associated with warm clouds that occur regularly in the rainy season in the east of the NEB, where SEALBA is located, MERGE has a vast network of surface

observations, mainly from state meteorology centers. These centers maintain and measure precipitation in a large number of municipalities, sometimes with more than one rain gauge per municipality. These information bases are integrated by MERGE, reducing its dependence on good satellite estimates in this region. This is also an explanation for the lower effectiveness of CHIRPS at SEALBA, which only relies on information from INMET, and especially for ERA5 and ERA5Land, which do not have such sources and have judicious quality control systems that often disregard scarce measures, as well as series with a significant number of failures.

ERA5Land [29] has three major sources of observations for its reanalysis, the first is based on data from different satellites from different international agencies, the second is data from stations provided by the WMO, and the third is information on observed snow and in SYNOP codes and coming from NOAA/NESDIS IMS. (NOAA National Environmental Satellite, Data, and Information Service Interactive Multi-Sensor Snow and Ice Mapping System). ERA5Land includes a number of improvements over ERA5, and this may explain the better dexterity scores over ERA5. Covering only variables over the Earth's surface, ERA5Land includes finer interpolation in time and space, an additional sea level adjustment of the meteorological fields, as well as more efficient possibilities for the import of updates [29,76]. ERA5Land does not directly assimilate data from observations but does so implicitly through atmospheric fields assimilated from ERA5 [77]. It is not possible, however, to infer direct effects of topography according to the results of this study, given that four of the stations analyzed have altitudes close to mean sea level, and the other three do not exceed 300 m. However, it can be inferred that the ERA5Land reanalysis may have improved the estimation of precipitation from breeze systems, which has always been a complicating factor for rainfall estimation in other reanalyses, by presenting more reliable adjustments of meteorological fields to the sea level.

### 3.4. Susceptibility to Droughts

As already shown, the March–August semester, which comprises the autumn-winter, is the wettest, and short-cycle crops, such as beans, corn, rice, cotton, and soybeans should be cultivated during this period, which can yield up to two crops in rainfed regime depending on the crop. However, despite not directly belonging to the Brazilian semi-arid region, SEALBA suffers recurrently from droughts, which among the adverse natural phenomena, is the one that most affects society, as it compromises large territorial extensions and acts for long periods [78].

Drought is characterized as a sufficiently long period of water deficit that significantly impacts edaphic, meteorological, agricultural, hydrological, and social aspects [79]. This drought occurs mainly due to the deficit of precipitation; however, it is intensified in the NEB region by the high amounts of evapotranspiration [80]. It is a phenomenon that imposes difficulties in the characterization of properties such as spatio-temporal beginning and extension, which makes monitoring difficult [81]. More recently, the 2012–2016 drought was the longest and most severe ever recorded in recent decades in the NEB, which intensified the interest of the scientific community in studying the subject [82–89].

In relation to our results, it is shown that in SEALBA this drought was longer, from 2012 to 2018. Figure 6 shows the precipitation deviations for the March–August semester in the last 12 years, between 2010 and 2021. In 2010, the rains were slightly above normal and in 2011, most of the region had normal precipitation, with the exception of the drier south and the wettest north. Between 2012 and 2018, precipitation was below average, with emphasis on the sharp negative deviations in 2012 (Figure 6c), 2013 (Figure 6d, with the exception of the north above average), 2014 (Figure 6e), 2015 (Figure 6f), 2016 (Figure 6g), and 2018 (Figure 6i) throughout the SEALBA, more concentrated in the center-south of the region in 2013 (Figure 6d) and 2014 (Figure 6e). In 2017 (Figure 6h), precipitation was close to average, with negative deviation cores further north and south as well as positive deviations in the extreme north and central portion of SEALBA. In 2019 (Figure 6j), negative deviations were concentrated in the north of the region, and wider positive deviations were

again observed in the central part. In 2020 (Figure 6k), deviations within the normal range predominated, with the exception of the central portion slightly above the average, and in 2021 (Figure 6l), negative deviations returned in the center-south of the SEALBA.

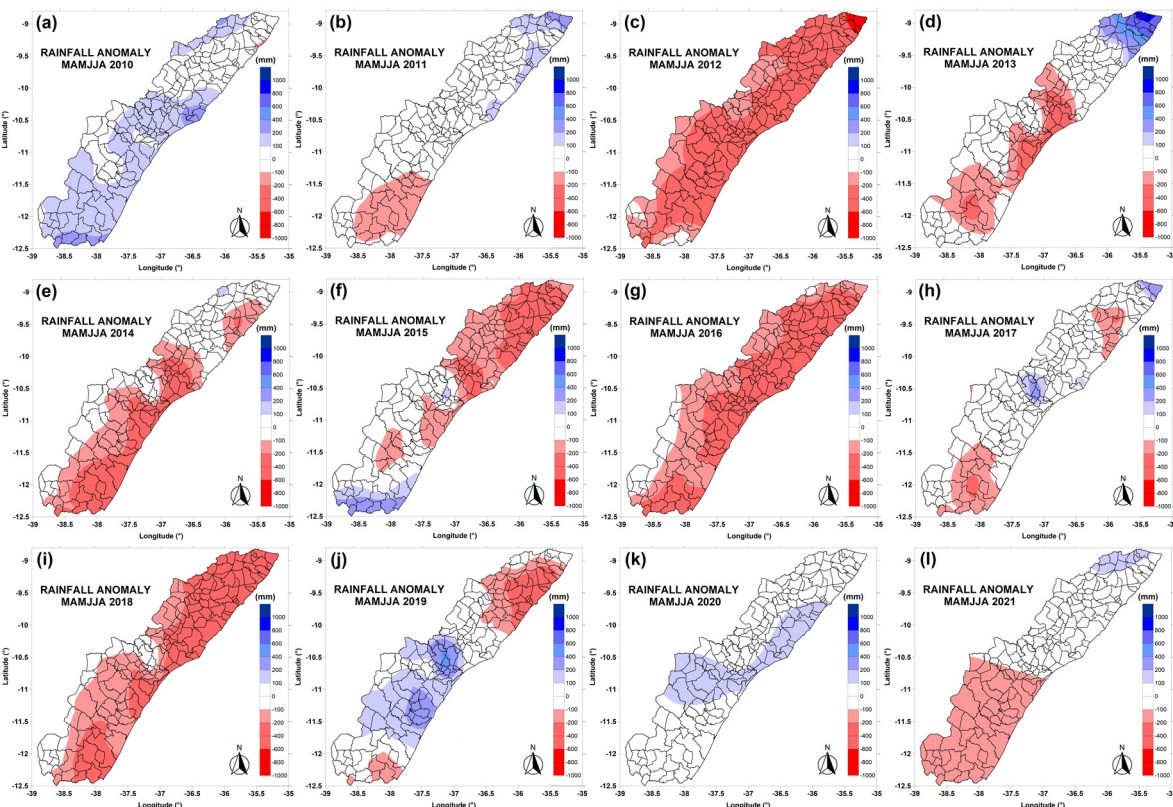

**Figure 6.** Precipitation anomalies for the rainy semester (MAMJJA) in the SEALBA region, for each year in the 2010–2021 period (which is also the reference period for the calculation of the anomalies). Top row: 2010–2013 (**a**–**d**); Middle row: 2014–2017 (**e**–**h**); Bottom row: 2018–2021 (**i**–**l**).

In a detailed analysis of the factors that led to the greatest drought recorded in the São Francisco River basin, the largest in extension and volume of water in the NEB [90], also concluded that the upper portion of this basin experienced a prolonged drought from 2012 to 2018. However, the area of this basin that involves most of the SEALBA was partially analyzed in this study. In a recent study [91], showed that in the period 1961–2017, the east of the NEB, including SEALBA, was the area that presented the most severe droughts, with an average duration between 14 and 24 months.

## 4. Concluding Remarks

SEALBA emerges with high potential to become the newest Brazilian agricultural frontier of grain production, since field research has already shown, for example, that soybean productivity is considered from high to very high.

With only seven long-term weather stations throughout its territory, this study showed that precipitation can be obtained for climatic and agrometeorological studies from four main sources: MERGE, CHIRPS, ERA5, and ERA5Land. Among these, MERGE, which is an improvement of the precipitation data estimated by the GPM-IMERG-EARLY satellite from the combination with surface observations, showed the highest correlations and lowest errors, followed by CHIRPS, ERA5Land, and ERA5.

Our results showed that none of the data sources should be completely excluded and in the absence of information from one of them, the others can and should be consulted, respecting the MERGE, CHIRPS, ERA5Land, and ERA5 hierarchy.

Despite being a very specific area in the east of the NEB, these results prove that it is very likely that the ERA5Land data represented a significant improvement in the estimation

of surface precipitation of the ERA5, from which it was derived. Additionally, ECMWF reanalyses have some advantages over MERGE and CHIRPS; since these data sources only produce surface precipitation data, reanalyses provide data on many other variables, both at the surface and at different atmospheric levels, which can be easily combined with better quality rainfall data from MERGE and CHIRPS, for example, as input to agrometeorological models for crop yield estimation.

**Author Contributions:** Conceptualization, E.H.d.L.S., F.D.d.S.S. and R.S.d.S.J.; methodology, E.H.d.L.S., F.D.d.S.S. and R.S.d.S.J.; software, E.H.d.L.S., R.S.d.S.J., I.G.F.d.F. and F.D.d.S.S.; validation, E.H.d.L.S., R.S.d.S.J. and F.D.d.S.S.; formal analysis, F.D.d.S.S., D.D.C.P., R.L.C., H.B.G., J.B.C.J. and D.L.H.; data curation, E.H.d.L.S., F.D.d.S.S. and R.S.d.S.J.; writing—original draft preparation, E.H.d.L.S., F.D.d.S.S. and R.S.d.S.J.; writing—review and editing, R.L.C., D.D.C.P., I.G.F.d.F. and J.B.C.J.; visualization, H.B.G.; funding acquisition, F.D.d.S.S., R.L.C. and D.L.H. All authors have read and agreed to the published version of the manuscript.

**Funding:** This work was partially funded by the following project of the Coordination for the Improvement of Higher Education Personnel (CAPES, acronym in Portuguese): CAPES/Modelagem#88881.148662/2017-01.

**Data Availability Statement:** The observational data used in this study is made available by INMET through their Meteorological Database ("Banco de Dados Meteorológicos do INMET"), at https://bdmep.inmet.gov.br/ (accessed on 1 February 2022).

**Conflicts of Interest:** The authors declare no conflict of interest.

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
