# Peer review of "Performance Assessment of Different Precipitation Databases (Gridded Analyses and Reanalyses) for the New Brazilian Agricultural Frontier: SEALBA"

_water, doi:10.3390/w14091473_

Round 1

Reviewer 1 Report

The article is well designed aplaying simple statistical methods. Nevertheless, the Introduction gives rather the background to the development of Brazilian agriculture in recent years than to the subject of the study – the assessment of different precipitation databases.

Here are a few notices which should be considered:

-Aracaju station on the south coast shows considerably lower precipitation than the other two coastal stations. This trend is not expressed in stations in the „transition area“. Is there any explanation for this fact?

-Only 7 surface stations were used in the evaluation of 4 databases but a vast network of surface observations in NEB is mentioned in connection with MERGE development [17]. Was there any reason for using only 7 stations?

-Extensive references do not always refer to the point. For example [9] does not involve precipitation analysis (rows 69-70). References should be checked once more.

Author Response

Replies to Reviewer 1

Dear reviewer, first, I would like to thank you for your availability to review our manuscript and for the valuable suggestions you provided to improve our work. Next, we answer each of them point by point.

The article is well designed aplaying simple statistical methods. Nevertheless, the Introduction gives rather the background to the development of Brazilian agriculture in recent years than to the subject of the study – the assessment of different precipitation databases.

Answer: Thank you for the positive comment on the manuscript. This fact was also noticed by other reviewers, and thus we even added a paragraph at the end of the Introduction, justifying the need for this type of research, right after the paragraph in which we cited the objective, to emphasize its importance. We really highlighted the historical evolution of agriculture in Brazil because this research is an initial step for projects to develop this activity in this new agricultural frontier.

  1. Aracaju station on the south coast shows considerably lower precipitation than the other two coastal stations. This trend is not expressed in stations in the „transition area“. Is there any explanation for this fact?

Answer: Indeed this is a fact, and it may have some answers. The first and most important is that Aracajú would be in a “transition zone” in the east of the Brazilian Northeast, in the following sense: it does not suffer the same influence of rains coming from frontal systems that manage to reach, in most cases, the coast of Bahia. (we cite an article in the text that deals with this theme), despite concentrating the rains from May to July due to the influence mainly of eastern wave disturbances (EWDs), other areas further north, such as the cities of Maceió and Porto de Pedras, seem to have the weather conditions influenced by this system in a greater frequency and intensity (we cite articles in the text that deal with this topic), and finally, as it is a relatively recent period of the last twenty years, some extreme events located of very extreme rains, such as those that occurred on the coast of Alagoas in the autumn/winter of 2010 and 2017 must have been decisive for the increase of their average precipitation in the rainy season in relation to Aracajú, which suffered from fewer extreme cases of precipitation accumulation in the same period.

  1. Only 7 surface stations were used in the evaluation of 4 databases but a vast network of surface observations in NEB is mentioned in connection with MERGE development [17]. Was there any reason for using only 7 stations?

Answer: The main reason is the public and always available nature of these data, and because they are from INMET, we are sure of their quality control. Of course, it is an advantage for MERGE to have access to data from local rainfall networks, but which unfortunately are not available to all sectors that need them, such as research and production. INPE, as it has a specific agreement with state Meteorology agencies, maintains its filing and processing internally to generate the MERGE. So the difficult access to such data, a tradition that should not change soon, accredits INMET stations for this type of analysis presented in our manuscript.

  1. Extensive references do not always refer to the point. For example [9] does not involve precipitation analysis (rows 69-70). References should be checked once more.

Answer: Thank you very much for noticing and alerting us. This reference was not in its correct position, so we corrected it in this revised version of the manuscript. Furthermore, we check the consistency of each citation, including changing the position of some and inserting new ones according to suggestions from other reviewers.

We appreciate all your questions, they were very important to our review process.

Reviewer 2 Report

Precipitation is very important for plant production (e.g. Wang et al., 2022, Frontiers in Plant Science; Yu et al., 2019,Science of the Total Environment; Shen et al., 2014, Remote Sensing; Wang et al., Polish Journal of Ecology), even if in some alpine regions (e.g. Tibetan Plateau) where temperature is always to be treated as limited factor. Therefore, this study has certain research value and significance. The following modifications need to be made before the article is accepted: 

  1. Since water is an international journal, the authors should not only mention the importance of precipitation in Brazil. They should also underline the importance of precipitation outside the Brazil. For example, many previous studies have found that precipitation is also important on the Tibetan Plateau(e.g. Wang et al., 2022, Frontiers in Plant Science; Yu et al., 2019,Science of the Total Environment; Shen et al., 2014, Remote Sensing; Wang et al., Polish Journal of Ecology; Fu et al., 2018, Agricultural and Forest Meteorology). Therefore, I suggested the authors made some related revisions and added some related contents.
  2. Please give numerical results in the summary section.
  3. Please refer to the software used in drawing and analysis.
  4. Besides RMSE, I suggested that the authors should consider other parameters, such as bias and MAE (Fu et al., 2011, Acta Ecologica Sinica; Fu et al., 2017, Journal of Resources and Ecology; Shen et al., Journal of Resources and Ecology; Wu et al., 2018, Remote Sensing Letters), to compare the accuracy among different datasets.
  5. The figure is not beautiful. The font of some figures is too small to see clearly, especially the abscissa and ordinate axes should be prominent, such as those in Figure 2 and figure 3.
  6. In Figure 6, please mark the serial number a.b.c. of the subgraph, At the bottom of the picture, the meaning of the facial beauty picture is explained.
  7. Figure 4-6 lacks the direction and unit of scale and abscissa and ordinate, please supplement.
  8. All saliency symbols should be italicized. For example, the p value of "p-value" should be italicized.
  9. The overall structure of the paper is better to separate the result part from the discussion part, and the discussion part analyzes the main conclusions. The current framework does not see what your main conclusion is.
  10. The conclusion part is wordy. The conclusion part only needs to give the summary results and prospects of the paper.

References:

  • Wang Jiangwei, et al. The change in environmental variables linked to climate change has a stronger effect on aboveground net primary productivity than does phenological change in alpine grasslands.  Frontiers in Plant Science, 2022, 12, doi: 10.3389/fpls.2021.798633.
  • Yu Chengqun, et al.  Effects of experimental warming and increased precipitation on soil respiration in an alpine meadow in the Northern Tibetan Plateau. Science of the Total Environment, 2019, 647: 1490-1497.
  • Fu Gang, et al.Increased precipitation has stronger effects on plant production of an alpine meadow than does experimental warming in the Northern Tibetan Plateau.  Agricultural and Forest Meteorology, 2018, 249:11-21. 
  • Wang Shaohua, et al. Interannual variation of the growing season maximum normalized difference vegetation index, MNDVI, and its relationship with climatic factors on the Tibetan Plateau. Polish Journal of Ecology, 2015, 63 (3): 424-439.
  • Shen Zhenxi, et al. Relationship between the growing season maximum enhanced vegetation index and climatic factors on the Tibetan Plateau. Remote Sensing, 2014, 6 (8): 6765-6789.  
  • Wu Jianshuang, et al.Modelling aboveground biomass using MODIS FPAR/LAI data in alpine grasslands of the Northern Tibetan Plateau. Remote Sensing Letters, 2018, 9(2): 150-159.  
  • Shen Zhenxi,et al. Estimation of daily vapor pressure deficit using MODIS potential evapotranspiration on the Tibetan Plateau.  Journal of Resources and Ecology, 2018, 9(5): 538-544.
  • Fu Gang, et al. Modeling aboveground biomass using MODIS images and climatic data in grasslands on the Tibetan Plateau. Journal of Resources and Ecology, 2017, 8(1): 42-49.
  • Fu Gang, et al. Estimating air temperature of an alpine meadow on the Northern Tibetan Plateau using MODIS land surface temperature.  Acta Ecologica Sinica 2011, 31:8-13.

Author Response

Replies to Reviewer 2

Dear reviewer, first, I would like to thank you for your availability to review our manuscript and for the valuable suggestions you provided to improve our work. Next, we answer each of them point by point.

      1. Since water is an international journal, the authors should not only mention the importance of precipitation in Brazil. They should also underline the importance of precipitation outside the Brazil. For example, many previous studies have found that precipitation is also important on the Tibetan Plateau(e.g. Wang et al., 2022, Frontiers in Plant Science; Yu et al., 2019,Science of the Total Environment; Shen et al., 2014, Remote Sensing; Wang et al., Polish Journal of Ecology; Fu et al., 2018, Agricultural and Forest Meteorology). Therefore, I suggested the authors made some related revisions and added some related contents.

Answer: We fully agree with your comment and take the opportunity to insert the following sentence in the Introduction, which is relevant and in line with the development of our introductory text: “Precipitation is the main limiting factor for the success/failure of production, directly responsible for the increase in soil moisture levels, impacting all arable areas on the planet, which can be monitored by indices that express vegetative vigor (Shen et al, 2014; Wang et al, 2015; Fu et al, 2018; Yu et al, 2019; ; Wang et al, 2022)”. In this way, we were able to insert the valuable references that were indicated to us.

  1. Please give numerical results in the summary section.

Answer: We did this by adding to the summary the average results obtained from comparing each analysis/reanalysis with the observations.

  1. Please refer to the software used in drawing and analysis.

Answer: We added this information in section 2.4, stating that all computational analyzes were performed using R software in its version 4.0.3.

  1. Besides RMSE, I suggested that the authors should consider other parameters, such as bias and MAE (Fu et al., 2011, Acta Ecologica Sinica; Fu et al., 2017, Journal of Resources and Ecology; Shen et al., Journal of Resources and Ecology; Wu et al., 2018, Remote Sensing Letters), to compare the accuracy among different datasets.

Answer: We appreciate this recommendation, and we have calculated the suggested indices. However, reading the suggested articles (and which we have inserted as references), among others, we realized that it would be redundant to add bias and MAE in the analyses, since the maps in Figure 5 already show us, in a way, the bias of the analyses/reanalyses when taking the MERGE as the closest measure to the observations. In addition, analyzing the suggested articles, we noticed that most of the time the authors prefer to show either bias and RMSE, or MAE and RMSE. Furthermore, our decision to add only the MAE was also due to the fact that the bias and the MAE differ only by the use/non-use of the absolute values of the differences between (re)analysis and observation, and the fact that, when the differences are totally positive, the bias and MAE values will match.

  1. The figure is not beautiful. The font of some figures is too small to see clearly, especially the abscissa and ordinate axes should be prominent, such as those in Figure 2 and figure 3.

Answer: We reconstructed the figures in order to highlight the parts that were not easily visible, especially on the axes.

  1. In Figure 6, please mark the serial number a.b.c. of the subgraph, At the bottom of the picture, the meaning of the facial beauty picture is explained.

Answer: We put the indications in the figure as suggested. In the text, when we relate to them, we use the indications of the letters so that the reader can easily find them.

  1. Figure 4-6 lacks the direction and unit of scale and abscissa and ordinate, please supplement.

Answer: We complete as suggested by informing in the Figure legend that P refers to the average monthly precipitation accumulated in Figure 4, and the respective differences in Figure 5, and we insert the direction in Figure 6.

  1. All saliency symbols should be italicized. For example, the p value of "p-value" should be italicized.

Answer: We revised the entire text and are now all in italics.

  1. The overall structure of the paper is better to separate the result part from the discussion part, and the discussion part analyzes the main conclusions. The current framework does not see what your main conclusion is.

Answer: Dear reviewer, we appreciate your suggestion, however we decided to keep the current structure of discussion associated with the presentation of the results. We justify this because, in addition to the main objective of the research, which was to list the best data sources in terms of skill, and which we believe we have made clear with the presentation of the main results of intercomparison between databases and observations, we seek to show other results such as the comparison of a rainy and a less rainy weather station and also the susceptibility of the area to the recurrent droughts of the NEB. Therefore, we kept each result presentation immediately associated with its discussion and comparison with other results found in the literature, leaving to improve the conclusions more explicitly, making it more direct in relation to the main points found.

  1. The conclusion part is wordy. The conclusion part only needs to give the summary results and prospects of the paper.

Answer: Dear reviewer, we removed additional information that really, when reviewing, is not necessary to be presented in this section, and we believe that we managed to make it less prolixe and more objective, with only the summarized results and the perspectives of the work, as suggested.

----------------------------------------------------------------------------------------

NOTE: We apologize for a mistake we found in the original manuscript: some RMSE values assigned to the data sources were inverted. We detected and corrected this error, fixed the table and the presentation of results and discussions were also corrected when referring to such values.

Reviewer 3 Report

The theme of the article is interesting and well described.

Perhaps a slightly more detailed explanation of the differences in the four datasets used, two from observations and two from weather models so they are estimates, would be useful. In fact, the models, especially in the representation of precipitation, have limitations that perhaps could be better explained.

On page 4 line 122 there is a citation different from the other, Huffman et al., 2015 respect to a number in square and it is not present in the references.

Author Response

Replies to Reviewer 3

Dear reviewer, we appreciate your words of appreciation regarding our article.

We made several changes throughout the text, in the abstract we brought numerical results according to the suggestion of another reviewer, and also in the Introduction when we mainly reinforced the main motivation/justification of the manuscript, and especially in the results and discussions section, where we brought more evidence the differences and similarities between the databases, and the conclusions, which were rewritten in a less prolixe and more focused on the main results of the research. We redid and adjusted figures, according to other reviewers' suggestions and verified and corrected the calculation of some dexterity indices presented in Table 2, which brought a reformulation of the analysis of the results and the discussions about the data sources. We took advantage of and inserted and updated all references in the article.

Thank you very much for your considerations.

Reviewer 4 Report

At the first, I would like to thank the Authors for the interesting manuscript. The aim of the work was the assessment of different precipitation databases. The Authors analyzed two gridded and two reanalyses. I have read the manuscript very carefully and in my opinion, it is prepared very well. The Authors have focused on the simple comparison using criteria such as RMSE and Pearson correlation. Indeed in my opinion the scientificity of the work is not high but I believe that similar studies are very important for particular practitioners. Currently, we are observing the less and less number of meteorological stations around the world. Thanks to similar studies we can indicate the most adequate substitute source of meteorological data if we do not dispose of the observed time series. In general, I believe that the presented manuscript should behave the chance to publish in Water, after minor comments:
1. In the Introduction must be clearly highlighted what is motivation and novelty of conducted work. 
2. Study area should provide more precise information, including land use and meteorological characterizations.
3. Table 2 in Results expect the statistical signification the Authors should provide information about the power of correlation. 
4. In the conclusion the Authors cannot cite any references. The conclusions must reflect only findings obtained by conducted studies.   

Author Response

Replies to Reviewer 4

Dear reviewer, first, I would like to thank you for your availability to review our manuscript and for the valuable suggestions you provided to improve our work. Next, we answer each of them point by point.

  1. In the Introduction must be clearly highlighted what is motivation and novelty of conducted work.

Answer: We appreciate your suggestion and this was really needed. At the end of the Introduction, we insert the paragraph that draws attention to the need for our study:

“It is worth noting that Brazil continually suffers from the gradual closure of weather stations [30], and the comparison of observed data with data from gridded (re)analyses in order to assess their performance is a necessity not only in Brazil, but at a global level [31,32]. In view of the increasingly present perspective of conducting climate studies and generating products for decision-making based on these sources of information, the need for this study is justified once again for an area of ​​Brazil that may soon consolidate it as the world's largest producer of grains. Furthermore, it is intended that such analyzes help in the validation of products from the surface observation database of the Brazilian Global Atmospheric Model (BAM) [33], which is the atmospheric module of the Brazilian Earth System Model (BESM), vi -going to obtain a hybrid dynamic-statistical coupling for the observed surface data and perform adjustments in seasonal forecast products for Northeast Brazil (NEB), with an emphasis on SEALBA.”

  1. Study area should provide more precise information, including land use and meteorological characterizations.

Answer: We have redone the figure of the study area, now much more interesting and showing more clearly the location of the stations and the topography. Also in the text, we removed information from conclusions that were not relevant in that section, as suggested by another reviewer, and added to the description of the study area:

“Figure 1 shows the location of SEALBA and its area in relation to the Brazilian territorial extension. It is an area of low relief, with some mountain areas that exceed 400m in height. In most of the region, altitudes barely rise above mean sea level. SEALBA's rainy season is in autumn/winter, and has a large number of perennial/semiperennial rivers that bathe this portion of the eastern part of the NEB. SEALBA has 32% of its territory located on the border of the Brazilian semi-arid region, with lower rainfall than in the extreme east. Among the advantages of this region are the proximity of two ports, located in Alagoas and Sergipe, in addition to two other ports in Bahia, which brings advantages for the flow of agricultural production in this region.”

More detailed information on the precipitating systems that operate at SEALBA is discussed in section 3.1.

  1. Table 2 in Results expect the statistical signification the Authors should provide information about the power of correlation.

Answer: We corrected Table 2, added one more dexterity index: the MAE, following the suggestion of another reviewer, and added information on the statistical significance of correlations in the text and below the Table.

  1. In the conclusion the Authors cannot cite any references. The conclusions must reflect only findings obtained by conducted studies.

Answer: Dear reviewer, we removed additional information that really, when reviewing, is not necessary to be presented in this section, and we believe that we managed to make it less prolixe  and more objective, with only the summarized results and the perspectives of the work, as suggested.

----------------------------------------------------------------------------------------

NOTE: We apologize for a mistake we found in the original manuscript: some RMSE values assigned to the data sources were inverted. We detected and corrected this error, fixed the table and the presentation of results and discussions were also corrected when referring to such values.

Reviewer 5 Report

Dear authors,
I thank you for your manuscript that aims to assess the quality of openly available gridded datasets for precipitation estimation in the SEALBA region of Brazil. I see it as highly focused on practical reasons and the needs of the agriculture industry. Probably, your study is a part of a big research project focused on the assessment of possible directions in agriculture development in the respected region. However, I see this practical and industry-based focus also as a drawback for *scientific* study. While you pose a clear scientific research question in the Introduction as "... to identify the best precipitation database for SEALBA among four available products", the answer to this question becomes clear (and, to be honest, trivial) after the introduction of datasets and methods. By the definition of their development workflows, MERGE and CHIRPS will be any way better if compared to meteorological stations because these products *assimilate* observations from these stations and also introduce calibration routines that try to minimize datasets' errors for known locations. Instead, ERA5 products do not implicitly assimilate precipitation measured on stations.

I understand, that the presented paper may be considered the first step to ensuring the usefulness of the domestic MERGE dataset for the SEALBA project frontiers. In this way, I think that you should focus more on the comparison of MERGE (as a domestic product) with station data and CHIRPS (as an international, or, US-based product), not with ERA5. To that, I recommend you to have a look at a methodology presented in Beck et al. (2017).

Beck, Hylke E., et al. "Global-scale evaluation of 22 precipitation datasets using gauge observations and hydrological modeling." Hydrology and Earth System Sciences 21.12 (2017): 6201-6217.

All the best and good luck.

Author Response

Replies to Reviewer 5

Dear reviewer, first, I would like to thank you for your availability to review our manuscript and for your valuable comments and improvements to our work. Next, we answer each of them point by point.

Answer: In fact, this work is the result of research aimed at promoting the expansion of a new frontier of agricultural grain production in Brazil, financed by the Brazilian Earth System Model (BESM) development project, which aims beyond improvement (calibration and validation) of its Brazilian Global Atmospheric Model (BAM) atmospheric module, applications in specific areas of interest such as SEALBA. In a good part of the SEALBA, sugar cane was produced and still is produced, historically. However, many spaces dedicated to this culture are now occupied by pastures or abandoned agricultural areas. Field experiments have shown that, with some soil management, most of these areas have high potential for the production of grains, such as soybeans (mainly) and corn. However, understanding the spatio-temporal variability of rainfall is crucial to determine the best agricultural calendars for such crops. Therefore, there is a real need to verify the best data sources for surface meteorological variables, with precipitation being the focus of this initial study.

There was a mistake in the initial manuscript that we detected already before the review process, but only after submission. We refer to an exchange of RMSE values ​​between ERA5 and ERA5Land, and also to an error in the calculation of the ERA5Land correlations with the observations. In this sense, now with everything corrected, we emphasize that MERGE and CHIRPS are still the best sources for precipitation, but ERA5Land, mainly, presented correlations and dexterity indices similar to those obtained with CHIRPS. In this sense, we discuss in the text the possible reasons that ERA5Land has significantly improved the results of ERA5, from which it was derived, and we present the advantages of being able to rely on the data from these reanalyses, which is a greater variety of data in addition to precipitation, such as the evapotranspiration, a fundamental variable as an input in several productivity estimation models (we emphasize in the text that the ERA5 data also presented correlations with high statistical significance, although greater errors than the other data sources, but which does not discredit it for use in the absence of information from other sources).

The article recommended for reading is great, we already knew it but after its indication we realized the importance of citing it in the text.

Thus, we conclude that the comparison of the four databases can bring us more benefits than a comparison only between MERGE and CHIRPS, and we greatly appreciate your comments that deepened our reflection on the results found.

Round 2

Reviewer 5 Report

I do not think that the authors put a considerable effort to revise the manuscript. Also, it is a pity to read that they "realized the importance of citing it in the text" rather than using it [article by Beck et al.] as guidance for the revision.

Anyway, I think that the authors follow the modern way of doing scientific work, and their study could be of course published as soon the major part of the community agrees on that modern way.